# What Can We Learn from Planning Instruments in Flood Prevention? Comparative Illustration to Highlight the Challenges of Governance in Europe

**Mathilde Gralepois**

Department of Planning and Environment, Engineering School, University of Tours,
35 allée Ferdinand de Lesseps, 37200 Tours, France; mathilde.gralepois@univ-tours.fr; Tel.: +33-247-361-461

**Abstract:** Studying the selection of planning instruments in flood prevention can be critical to gain a better understanding of governance. This choice is underestimated in the flood management literature. This paper fills a knowledge gap in flood management governance by examining the rationales for the choice of instruments. The study is grounded on a comparative illustration of planning instruments in flood prevention in three European countries: England, France and the Netherlands. Flood prevention through spatial planning is a specific example, as the implementation of the Floods Directive has reactivated the role of spatial planning in urban agglomerations. The choice of instruments is never neutral. In the field of flood management, alignment among strategies is supposed to lead to resilience. Instruments should be aligned and coherent. Is that the case? The article explains the challenges of governance configured by a conflict between the spatial planning policy steered by local authorities and the risk prevention policy led by national authorities. This model is further complicated by the tension between the preference for legal, technical or scientific instruments, and the difference in professional culture between planning and prevention. The selection of instrument shows that if their conflicts are exacerbated to debates on variables or parameters, it is because there is no political agreement on the balance between development and security.

**Keywords:** flood prevention; policy instruments; spatial planning; governance; resilience

## 1. Introduction

Since the 1960s, flood management has been dominated by hazard assessment, defense strategy and infrastructural measures, as if human societies could indefinitely control nature [1,2]. In the 1990s, extreme hydrological events occurred in the River Rhine (1993, 1995), in the Mediterranean region (1994), and in Central Europe (1997) [3]. After this flood-rich decade in Europe, policymakers have also focused attention on flood prevention and crisis management [4,5]. More precisely, flood management policy now tends to diversify its strategies in Europe [6]. Despite political and institutional efforts of diversification, flood risks persist, and disasters cause increased damage. Since 2000, floods in Europe have caused at least 700 deaths, and €25 billion in insured economic losses [7]. This trend is not decreasing. Several studies and reports underline that global warming increases the frequency of river floods in Europe [8–10].

During the 2000s, flood risk management at the European level extended the scope of interests in flood governance [2,11]. The consequences of natural disasters are considered as management failures, not only flood hazards [12]. The difference is radical: flood management shifts from controlling nature to adapting to uncertainty: "Risk is typically viewed as something that can be described in statistical terms, while uncertainty is viewed as something that applies to situations in which potential outcomes and causal forces are not fully understood" [13].

Flood management is not a question of reduction or mitigation, but a challenge of adaptation and transformation. This challenge is not only a question of improving parameters, factors and models but of integrating the uncertainty associated with each input variable [14]. Rather than necessarily being a barrier, uncertainty is an opportunity to reconsider risk governance and to better handle floods [15].

The literature on flood governance argues for a diversification of flood management strategies, first to respect the objective of the Floods Directive [16], but more broadly to promote the implementation of resilience [17]. Five types of flood risk strategies can be cited: action on probability of flooding (Flood Defense); on the potential consequences of flooding (Flood Prevention, Flood Mitigation, Flood Preparation) and on recovery after a flood has struck (Flood Recovery) [2]. Beyond the objectives for the diversification of the different policy strategies, coordination and alignment of these strategies is a prerequisite [18]. This comprehensive approach is difficult to implement concretely in flood governance in Europe. The literature explains how path dependence on institutional routines, traditional political arrangements or traditional technical choices are barriers to more resilient and adaptive governance [1,11,19]. Nevertheless, there is no radical change in European public policy, national levels or local implementation. To take a step forward, flood governance literature explores precise issues, such as coordination mechanisms [20] and science-policy interfaces [21]. This paper believes that those precise inputs are likely to contain general insights and should be pursued. It is argued in this paper that policy instruments are underestimated in flood governance literature [22], for example, in the selection of instruments of spatial planning in flood prevention.

Flood prevention can be described as an attempt to avoid the negative consequences of flooding mainly by means of proactive spatial planning ("keeping people away from water"). It aims at building away from areas prone to flooding by applying rules banning construction, expropriation or, more often, measures to adapt buildings to flooding. Public information could also be cited in flood prevention instruments, but there is not much evidence to measure their efficacy because few policy dynamics are observed in the information instruments [22–24]. By contrast, the production of planning instruments is dynamic in Europe. The implementation of the Floods Directive reactivates the role of spatial control instruments, especially in concentrated urban settlements. Growing urbanization brings about major challenges regarding aggravating flood risk. It reduces the capacity of cities to resist and to adapt to floods by reducing the capacity for infiltration, evaporation and absorption, as well as increasing the concentration of populations, activities, services and networks. This both increases the likelihood of flooding and the vulnerability of urban contexts [5,25–28]. In addition, flood regimes are fostered by other human activities such as intensive agriculture, deforestation or motorized traffic [29,30], especially in the context of climate change [8].

The paper aims to address the lack of knowledge, both for practitioners and academics, on the use and limits of planning instruments for flood prevention from a European perspective. It fills a vacuum in risk management by firstly examining the rationales for the selection of planning instruments, which today is critical to gain a better understanding of governance. The following section considers the conceptual insights of policy instruments for a better understanding of flood governance. Then, a third section provides an overview of the empirical dataset used to draw the comparative illustration between England, France and the Netherlands. Results on what the place of planning instruments in flood prevention reveals for a better understanding of flood governance in the context of urban resilience are presented in the fourth section. The final section draws conclusions and reflects on future directions.

## 2. Conceptual Insights of the Choice of Instruments for Flood Governance

This paper proceeds from the observation that there is a lack of studies on the specific role of instruments in flood governance. Although there have been studies on the complex interactions between urban planning and flood prevention, there has been no study of instruments.

## 2.1. The Role of the Choice of Instruments in Governance Studies

The paper studies the planning of policy instruments for flood prevention in European cities. If flood management is composed of several strategies, each strategy is implemented through instruments. These instruments matter. A strategy can be defined as a combination of the measures and the resources necessary to implement them. Instruments stand as a means of application, involving a certain use of resources and techniques to attain a set of goals within the framework of a specific strategy [31]. Put differently, instrumentation is "the set of problems posed by the choice and the use of instruments (techniques, methods of operation, devices … ) that allow governmental policies to be made material and operational" [32] (p. 4). Policy scholars agree that the selection of instruments is never neutral. Indeed, among the number of instruments available in the public policy toolkit, there is room for choice [33,34]. This choice reveals modes of governance. However simple it might be, instruments are key to understanding trends in governance [31,35,36]. The parameters, justifications and applicability of an instrument are not secondary issues. Depending on its own structure and logic, every instrument constitutes a condensed form of power in governance.

Although research on policy instruments has been prolific in recent years [37], it has received little attention in the field of flood management [22,38]. Our goal is to offer an overview of governance trends in flood prevention in Europe through the role of planning instruments. The paper starts with a conceptual framework inspired both by the instrument analysis of Le Galès and Lascoumes [32] and the governance modes from Howlett [31].

Lascoumes and Le Galès analyze policy instruments through a multidimensional typology, highlighting the links between instruments, legitimacy and political relations. Policy instruments reveal types of public policy outcomes "in [their] meaning [and] in the cognitive and normative framework" [32] (p. 16). Policy instrumentation reflects specific governance choices and the ways in which they evolve. Elaborating on Hood's reference work [33], Lascoumes and Le Galès identify five types of instrument:

- Regulatory instruments: traditional coercive tools of state interventionism based on legal forms;
- Economic instruments: also based on legal dimensions, their peculiar feature is to resort to monetary techniques in a way either to redistribute resources or to redirect them;
- Incentives: in a critical context of bureaucracy, the rigidity of legislative and regulatory rules, these agreements coupled with sets of incentives have become a general injunction;
- De facto standards: made by governmental actors to organize power relations within civil society, they frame methods or conditions of production for services;
- Communication and information-based instruments: part of what is generally referred to as the "open" or "grassroot" democracy and participative public.

This typology is useful for describing public policies, and especially for studying the role of the state, its political relations and its types of legitimacy. As public authorities are central actors in flood management, especially in prevention strategy, the typology has a concrete application. Nevertheless, an earlier study on flood management instruments [22] reveals that "soft instruments", such as best practices, incentives or communication, have not really permeated European flood management. More generally, the three last categories overlapped, as few examples already exist. Somehow, it underlines the lack of co-production in flood management and the need for adjustments within civil society [24].

Working on instruments, Howlett proposes a table of modes of governance [31]. Public policies are a multi-level and nested process. The choices of policy instrument are "all about constrained efforts to match goals and expectations both within and across categories of policy elements" [31] (p. 74). The calibration of instruments is tricky because it requires achieving several goals, in multiple policy levels, at the same time. Flood management is, in Europe, a fairly long-term and stable governance arrangement [2], grounded in modern liberal-democratic states. His typology is pragmatic, giving concrete examples of instruments. Howlett's typology offers four different modes of governance with

which to study the choice of instruments and explains how it reveals governance. As in Lascoumes and Le Galès' typology, Howlett identifies a "Legal Governance" and a "Market Governance".

Next, Howlett makes room for a corporatist mode of governance, which fits with the expertise and technocratic pattern of public instruments in flood prevention, such as plans, maps or models. He then merges the different trends of non-mandatory but competitive instruments, such as standards, best prices or charts, in a "network pattern". I propose to merge the two typologies in one table (Table 1).

**Table 1.** From the choice of instruments to the legitimacy of governance.

| Implementation Preference | Type of Instruments | Mode of Governance | Governance Aim | Type of Legitimacy |
|---|---|---|---|---|
| Legal system: legislative and regulatory preference | Legislation, rule, law, regulation. | Legal governance. | Legitimacy and compliance through the promotion of law and order. | Imposition of a general interest by mandated elected representatives. |
| Expertise system: macro-level bargaining | Model, map, plan, scheme, framework. | Corporatist governance. | Controlled and balanced socio-economic development through the management of major organized social actors. | Justification by socio-technical expertise from a top-down perspective. |
| Market system: economic and fiscal preference | Contracts, Subsidies, tax incentives, penalties. | Market Governance. | Resource/cost efficiency and control through the promotion of competition. | Benefit to the community through social and economic efficiency. |
| Network system: agreement-based mobilization | Collaboration, voluntary associational activity and service delivery. | Network governance. | Cooptation of dissent and self-organization of social actors through the promotion of inter-actors organizational activity. | Explanation of decisions, direct involvement and accountability of actors. |

Modified from Howlett [31] with some inputs from Lascoumes and Le Galès [32].

This table helped to guide the analysis of the paper, to explain the distribution of instruments, and to categorize and capture the evolution of flood governance.

## 2.2. Crossing Urban Planning and Flood Prevention through Instruments

European cities were considered as secure and resistant to floods. In the 1990s–2010s, dramatic and recurrent floods in cities in the United Kingdom, the Czech Republic, Bulgaria, Hungary, France, Romania and Spain constantly highlighted the emergency of the situation. Policymakers, as well as the general public, witnessed the vulnerability of cities and the need to prioritize proactive spatial planning. Prevention strategy in flood management mostly revolves around spatial planning. Its main objective is to influence the location of concrete activities as well as future spatial and economic developments. The main objective is to regulate urbanization by banning construction in flood-prone areas and allowing development under conditions elsewhere (upper-elevation, choice of building, materials).

Yet cities are specifically vulnerable to floods, as they are densely populated, with large numbers of activities, services and networks, with a limited capacity for discharge and absorption. They also experience continuous geographic and demographic urban development [39]. During the first step of the Floods Directive implementation, countries have identified cities as areas most at risk from significant flooding [40]. The Floods Directive requires EU Member States to implement flood maps and prevention plans for spatial planning decisions [7]. More than ten years after the Floods Directive was implemented, cities are still not adapted to flood risks [41].

Connecting planning, especially urban planning, and flood prevention received attention in the 2000s [42–48]. Analyzing the literature on urban planning and flood prevention reveals several obstacles in governance. During the years after World War II in Europe, urban planning overwhelmingly neglected the integration of flood prevention in cities. Urbanization has continued in flood prone areas. Even where flood land-use controls existed, they were underestimated, incomplete and depreciated in the face of the challenges of urban development [45–47]. In 2007, the Floods Directive required states

not only to formalize extreme flood scenarios, but to enter into the arena of spatial planning. It is difficult to coordinate the governances of water management and urban planning [44,49]. There are few incentives to develop a sense of risk-reduction "ownership" among urban planners. First, planners do not perceive risk management as part of their remit [49,50]. Secondly, significant differences can be observed in terms of historical and educational backgrounds between disaster-prevention and planning officials [51]. Spatial planning is supposed to be interactive and comprehensive [52]. Flood prevention is commonly implemented via expert-based hydraulic engineering of urban structures [1,11,53] including canalization of streams, construction of floodwater retention basins and diking.

While the governance literature studying urban planning and flood prevention highlights the divergences between priorities, interests, experiences, cultures or beliefs, the role of instruments is not analyzed. Moreover, separate studies focus on single countries, such as the United Kingdom [46,47], the Netherlands [52] or France [42,43,54,55]. This paper fills a knowledge gap with a comparative illustration of planning instruments in flood prevention strategy in different European countries: England, France and the Netherlands. In the field of flood management in Europe, alignment among strategies and coordination between scales (geographical, spatial, institutional) promotes urban resilience [6,18]. This paper looks at whether the selection of spatial planning instrument is aligned, coherent and convergent in flood prevention.

## 3. Methods

To address the question, the author examines the rationales for the selection of planning instruments in flood prevention. I proceed in three steps.

First, searching through national and international academic databases which cross-reference planning and risk management is an important starting point, especially given that the literature is not extensive. A total of 94 publications concerned with spatial planning in flood management were analyzed, mainly in peer-reviewed journals and book chapters from international scientific databases (Springer, Science Direct, Google Scholar, Wiley, Elsevier), as well as grey literature, such as expertise and public authority reports. These references are selected for their precise accuracy on the subject. They provide an updated and critical knowledge framework that effectively serves as a theoretical background.

Next, the empirical data were retrieved from two European research reports, as well as their minutes, briefs and notes: STARFLOOD and TRANSADAPT. These two research projects focused specifically on flood governance, although neither the intersections between risk and planning, nor the role of instruments, were central to the results. However, the quality of the data produced, and the diversity of the investigations carried out, enable the collection of a large quantity of data to be processed henceforth. I look at general insights from European flood prevention strategy as well as the specific role played by planning instruments. Concerning reports from the STARFLOOD database, the empirical data is based on country reports available to the public via the project reports of England [56], the Netherlands [57] and France [58]. Each country report is based on 50 interviews per country on average. Concerning reports from the TRANSADAPT database, between 20 and 30 interviews with national, regional and local stakeholders and citizens as well as practitioner workshops were organized in each country between May 2015 and March 2016. Data is collected in country reports for France [59], the Netherlands [60], and in a final comparative analysis [61]. In both reports, each interview was recorded and transcribed verbatim. Additionally, an analysis of the institutional context of hazard and risk management systems in each country was conducted, including an analysis of relevant policy documents. It built a substantial database including observations, policy and legal document analyses in which the issue of instruments was not fully studied. The qualitative data thus produced makes it possible to have access to the actors' representations of their interests and strategies, as well as the idea they have of those of other actors. By comparing these data with the results of the literature, I can analyze the choice of instruments, beyond their supposed neutrality.

Finally, three European countries were studied to draw a comparative illustration of urban planning instruments and evolution in flood prevention: England, France and the Netherlands, taking concrete examples from national and local instruments. "Comparative illustration" means that the document does not lead to a systematic term-by-term comparison. It draws insights from similarities and differences in the way each country used planning instruments to cope with flood prevention. The three countries present concrete differences in terms of their legal system, institutional structure, political traditions or administrative routines. But they are all rooted in the same cultural space of Northern Europe. Thus, this common social and societal context allows for a comparison.

## 4. Comparative Illustration of Planning Instruments in Flood Prevention to Highlight Governance Challenges in Europe

Analyses of the instruments used to face floods brings new understandings of flood prevention. Policy instrumentation may relate to the fact that actors find it easier to agree on methods than on objectives. While instruments are often seen as secondary administrative techniques, they have a huge potential to better understand governance choices.

### 4.1. England: The Legal Governance through Regulatory Instruments Remains Inadequate

In England, spatial planning instruments are organized with hierarchical articulations from national to local authorities. The National Planning Policy Framework sets out guidelines from the Ministry of Housing, Communities and Local Government on how flood risks should be incorporated into the overall planning system, with provisions regarding climate change. The National Planning Policy Framework provides top-down governance with the type of production, agenda, plan-making process and conditions of participation for Local Planning Authorities. Despite their prospective and strategic status, Local Plans have to incorporate national recommendations. Included in Development Plans, Local Plans provide specific allocations of land for different purposes. Local Plans are put into practice by Local Planning Authorities. They identify where, and how many, homes, buildings, businesses, shops, local infrastructures or networks should be located in respect of environment, sensitive landscapes, climate adaptation or flood prevention. Guidance for Local Plans have recently been incorporated into the guide on Plan-making (March 2019). "In addition to the statutory requirement to take the Framework into account in the preparation of Local Plans, there is a statutory duty on local planning authorities to include policies" [62]: Local Plans are documents with both strategic (objectives) and non-strategic (operational) policies. They should illustrate geographically the policies in the plan. Based on the Local Plans, Local Planning Authorities are responsible for "deciding whether a proposed development should be allowed to go ahead" through Land Planning Permissions [63]. Concretely, planning authorities have concern for flood prevention and mitigation. They bear the responsibility of assessing risks at the local level while ensuring that flooding is taken into account.

If building developments are planned in flood-prone areas, local authorities have a statutory requirement to consult the Environment Agency. To obtain a Land Planning Permission, any land developer is required to conduct a Flood Risk Assessment according to the rules set out in the Local Plans. This should contain the estimated flood level for the development, details of your flood resistance or resilience plans and any supporting plans and drawings [64]. A specific instrument of the English system is called the sequential test. "The sequential test compares your proposed site with other available sites to show which one has the lowest flood risk" [64]. Developers have to prove that they cannot build their project somewhere else. This mechanism aims at reducing developments in floodplains.

In sum, English planning instruments in flood prevention are implemented through a full legal system. Here, legitimacy is supposedly conferred by the common interest but is in fact a matter of socio-technical expertise imposed from on high.

At first, flood prevention was a matter of local bodies taking their lead from state doctrine. Nowadays, there is a trend towards managing floods at local scales (local authorities, communities) but still from a legal perspective [21,56]. The rules for spatial planning, such as Land Planning Permission, are supported by scientific hydraulic models and represented though geographic information mapping systems. The relations between planning and flood management are complex [45]. Flood management concerns are incorporated in planning systems and the legal strength of flood management has been reinforced during the 2000s [48]. However, this legal governance through regulatory instruments remains unsatisfactory if one considers the theoretical results and practical successes of citizen involvement [24,65,66]. Here, at the end, the selection of instruments remains unsatisfactory because spatial planning seems to be an intangible priority, whatever the conditions of environment, risk or uncertainty. Various options to veto developments are rarely used. Governance is supposed to be pyramidal from national recommendations to local permits, but Local Planning authorities are still permissive [48]. When national flood recommendations are not mandatory, they are ignored [47]. In a central mode of governance based entirely on legal instruments, Local Planning Authorities oppose other legal instruments stemming from public transport, social housing or energy policies to defend their own development interests. Beyond the legal system, there is no global agreement on the balance between development and prevention. This is evidenced by the fact that flood risk management is rarely a political issue in the sense that it is not addressed in political debates, in the media or in electoral campaigns outside specific natural crisis times [67,68].

### 4.2. France: Towards Resolving Controversies over Instruments through the Transfer of Competences?

In France, there is also a history of separation between spatial planning and flood management, coupled with a historical opposition between national and local authorities [69]. Dating back to the nineteenth century, flood prevention originated with the idea of floodplain preservation. The first flood planning laws were introduced in 1935. As of today, French flood prevention systems have diverged: the spatial planning system is led by local authorities, and risk prevention by the state. Concerning the planning system, land development is ruled by Spatial Strategic Plans (Schémas de Cohérence Territorial) as well as Local Urban Plans (Plan Local d'Urbanisme). The Local Urban Plan is enforceable against all private and public persons who want to build in this sector. Indeed, the Local Urban Plan is the more binding regulation. Concretely, building permits (Permis de Construire) are examined through the Local Urban Plan and then accepted or refused by local authorities on the basis of the French planning system, decentralized since the 1980s. To integrate flood prevention, one instrument is supposed to link the planning and flood management systems: the Flood Risk Prevention Plan (Plan de Prévention des Risques d'Inondation). Introduced in 1995, the Flood Risk Prevention Plan reinforces the state's responsibility in the flood management domain and the legitimacy of flood maps based on hydraulic models. It plays a key role in spatial planning, as part of a broader planning culture dominated by an engineered approach and legal restrictions on construction in risk areas [70–72]. Legally, the Flood Risk Prevention Plan is not a planning document, but a public easement adopted by the national authority that must be respected by all the building permits. The planning competencies of the local authorities are de facto restricted. For decades, the Flood Risk Prevention Plan has been a matter or a symbol of conflict, between land development and flood prevention, between the legitimacy of national or local authorities. The Flood Risk Prevention Plan is interpreted as a way for the central government to maintain control over both local authorities and the spatial planning decision process. Recently, changes have occurred: the transfer of competences of some flood prevention duties to local authorities. Justified by the Floods Directive implementation, but most probably to find a way out of decades of conflicts over Flood Risk Prevention Plans, a new competence for the management of aquatic environments and flood prevention (GEMAPI: Gestion des Milieux Aquatiques et Prévention des Inondations) was implemented in 2018. It allows local authorities to develop both their political legitimacy and their technical capacity [73].

In short, controversies over flood risk prevention in local plans highlight conflicts in governance. For some local authorities responsible for land development, unbuildable flood-prone areas represent a breaking point for their urban expansion programs, even more so when these local authorities have little free space left and experience a strong real-estate pressure. Their only way to continue building, and ensure economic development, is to tackle the issue of flood risk through the Flood Risk Prevention Plan. The flooding issue is a way for local powers to show their disapproval of central government trespassing on their own areas of competence and to progressively usurp their role in flood governance [25,74,75]. If the transfer of the legal instrument from national to local authorities can be seen as an opportunity to affirm themselves local governance, the new regulation GEMAPI represents an additional burden that will impact on already weak resources [68,76].

### 4.3. Netherlands: When Instruments are Mandatory but Non-Binding

In the Netherlands, "water" is considered the twenty-first-century challenge of spatial planning. More than 90% of the Dutch population lived in urbanized areas in 2018. If most of the residents are protected by the dike-rings system, 35% of all the inhabitants live in flood-prone area [77]. Many studies underline the need to strengthen the integration of land-use planning and flood risk prevention [52,78,79]. Strategic spatial development plans (structuurvisies) and legally binding spatial zoning plans (bestemmingsplannen) have to feature flood reduction provisions. They include specific bans or restrictions applicable to building work, as well as relevant expropriations or re-allocations. However, the rules are less embedded in spatial planning legislation than in England or France. Dutch flood prevention in spatial planning relies on two agreement-based incentives. Designed as bridging mechanisms, the Water Assessment (watertoets) is a formal advisory construction mechanism used to facilitate the integration of spatial planning and water management. Local planning authorities need to consult regional water authorities—which belong to Dutch water management—during the process of drafting spatial plans. The Water Assessment is mandatory to ensure that local plans fit the national criteria [80] but non-binding [81]. The watertoets practical implementation is described as ineffective for the prevention of inappropriate urban development or to support flood-proofing [78]. For example, the checks take place when the plans have been made, mostly almost finished. Other issues, such as economic growth, mobility, environmental quality, are prioritized.

Studies have shown that the lack of inclusion of risk prevention in urban planning offends flood managers because they prioritize the issue of safety and prevention over that of development [45,52,68]. Nevertheless, we must also side with the planners. Sometimes, the local authorities are obliged to follow old practices and traditional models, even if scientific developments have proven those practices and models inadequate. Instead of improving the flood prevention plan by considering urban development, how can flood prevention measures be integrated into planning instruments [82]? In brief, just as in England and France, legal instruments predominate. Furthermore, the Netherlands has a global strategy of diversification of flood management strategies through the concept of multi-layered safety [21,57]. The country makes a more diversified use of flood prevention instruments, especially via the Water Assessment, which is a pillar of the flood prevention system. Even though it helps to make flood management in spatial planning mainstream, this incentive-based instrument does not in practice systematically prevent constructions in flood-prone areas [80].

Considering the different modes of governance categorized by Howlett (Howlett 2009) and Lascoumes and Le Galès [32], flood prevention in planning is characterized by a classic pattern, rather resistant to change, of public actors, by hierarchical power relationships, whether from the national to the local level, or from the public authorities to civil society, and, finally, by compartmentalized public policies. Even if these observations have recently been acknowledged [11,52,83], the article draws new significant conclusions.

First, flood prevention in urban planning is largely based on legal instruments. England and France display particularly hierarchical legal structures in the field of flood prevention.

The Netherlands, even if it is not optimal, is diversified with incentive-based instruments. Governance is historically based on legislative preference, legitimated by the democratic representation of mandated representatives in public authorities. Nevertheless, the study of the instruments demonstrates that legal legitimacy is based more on a system of socio-technical expertise in a top-down perspective than on general interest.

Secondly, the comparative illustration of the policy instruments in three European countries confirms the solid role of hierarchical governance. Flood management has a long tradition of central governance: all water-related issues are described by Wiering and Crabbé as "hegemony of the state" [84] (p. 99). However, in the three countries, there is a shift toward devolving responsibilities to lower governments and the market [82]. When local authorities have flood prevention instruments, they are not binding and completely controlled by national authorities.

Thirdly, public policies in spatial planning and flood prevention are still working in parallel rather than together. They are not sufficiently complementary or interacting. There are many obstacles to future "Flood Risk Management Planning" [52]. There is currently no common practice concerning the introduction of flooding into development [82]. The next section examines how another step forward can be accomplished. For the time being, until the selection of instruments is adapted to the challenges of interdisciplinarity, neither policies nor governance can be integrated and resilient.

## 5. Discussion and Conclusion. A Step towards Integration of Flood Prevention and Spatial Planning by Looking at the Choice of Policy Instruments

Returning to the flood resilience literature, the case study analysis confirms that European governance face three challenges: adaptation, diversification and participation. First, flood governance requires a serious shift from resistance to adaptation of both physical and socio-economic systems [6,85]. In the three countries examined, protection remains a founding principle. The "stickiness" of the flood defense strategy can be explained by path dependency factors, especially with regard to instruments based on old practices and traditional models from national policies. However, when the position of resistance is shifting, changes are defined as part of a multi-layered and adaptive strategy rather than as a real change in nature [1]. Thus, the study of the power struggles between risk management and planning instruments shows that urban planners do not consider planning instruments to be a driving force in flood management. They consider that floods are controlled by protective measures elsewhere, such as in the Netherlands. Secondly, the diversification of several flood management strategies should be developed with multi-level coordination mechanisms in order to avoid unintended fragmentation effects [20,83]. Beyond the discourse on the alignment of strategies or the coherence of instruments, the case studies show that coordination between instruments must be anticipated at the early stages. Bringing together instruments developed separately, such as Risk Prevention Plans in France, takes decades of conflict before solutions are identified [86]. Thirdly, more bottom-up initiatives and powers for local authorities, inhabitants, communities and local business have to emerge to encourage collaborative public decisions and open public debate [24,61,65]. While there are mechanisms for debate on strategic orientations, citizen participation in the development of public policy instruments remains absent. In France, England and to a lesser extent in the Netherlands, the development and selection of instruments remains the hierarchical prerogative of experts. There are incentives to promote property adaptation in France and England. Also, in the Netherlands, local community forums help local policy practitioners and citizens to discuss and cooperate on measures to store rainwater on private grounds [24]. Nevertheless, these marginal points do not fully deal with planning instruments in flood risk areas. They provide resources to reduce the vulnerability of existing houses or they are part of a mitigation strategy concerning/aimed at rainwater management more than flood management.

The results of the case studies and the approach to instruments from a social science perspective provide new perspectives, both in literature and in practice. To address these three issues (adaptation,

diversification and participation) and because they are interconnected, the literature on flood governance could focus on two possible entry points.

The first possibility allows for an instrument-based approach by tackling the place of legal flood maps. The second approach is more general but offers the possibility of actor-centered or representation-centered approaches: it concerns the dilemma between development and flood prevention.

The first entry point, the implementation of the Floods Directive [16], promotes the application of legal maps to flood prevention [87]. Using a hydrologic-hydraulic approach and geographic information systems (GIS), flood maps have a statutory role in Local Plans [88,89]. As for the Flood Risk Prevention Plan in France, the use of planning instruments in flood prevention emphasizes that maps are not only geographical representations and indications, but an act of legal zoning to limit expansion and densification of construction. As socio-technical objects, flood maps are negotiated. Flood maps are subject to negotiation because the different methods lead to a completely different ranking of degrees of flood risk depending on the importance of the parameters [14,15,90]. Even with precise factors and data, it is impossible to draw definite conclusions on flood levels, magnitude, severity or probability [90]. Uncertainty is inherent in risk and allows—even requires—discussion, negotiation and selection. In France and England, the controversies are about procedures, scientific parameters, scales, even the "thickness of the line" that delineates the risk of flooding, and therefore the areas where you may or may not be able to build [70,71,74,88,89]. The negotiation of flood maps illustrates two trends in governance: the evolution of legitimacy and the evolution between professional cultures. The first trend, the negotiation of flood maps, shows the evolution of legitimacy between national and local authorities. The study of decision processes on flood maps illustrates the conflicts between the legitimacy of expertise and legal legitimacy. Until the 1980s, the national authorities in Europe possessed an unquestioned resource: expertise on flooding, i.e., the hazard assessment, the definition of the hydraulic model, the choice of scenarios and the return periods. With the weakening of the states in the 1990s due to the contradictory movements of internationalization and devolution, the legitimacy of central governments regarding expertise-based instruments has been challenged. Local authorities are contesting the scientific and technical resources to frame flood maps. Controversies over instruments have concrete implications for land development if local authorities are not implicated early on, especially for powerful European cities that have more resources than national authorities. Sometimes, for example, local authorities have access to more precise data, are more easily adapted in the new research developments or they can pay for alternative experts' reports to recalculate the scale of the flood map [74,88]. To deal with the local authorities' complaints, national authorities are focusing more narrowly on a strict application of legal and organizational rules (agenda, choices in the decision-making process, conditions of participation, etc.). It is a vicious circle, which could be broken by opening up the discussion to public participation and debate [65,91,92]. Nevertheless, several actors are little studied, especially private expertise actors. Governance literature lacks the analysis of private consulting bodies, working more and more for national and local authorities. Secondly, the negotiation of flood maps shows the evolution of governance between planning and flood management professional cultures. Governance is characterized by a double path: the spatial planning system led by local authorities, and the risk prevention system driven by the state. Hartmann and Driessen explain that water managers no longer defend the idea of "lines of defense" as a strict delimitation between an area at risk of flooding and one that is not [52]. This is despite their different backgrounds; planning has a holistic and multi-disciplinary approach and flood management is more specific and sectional [44]. If the boundaries are shifting, the study of instruments shows that planning and flood management are still two separate domains [82]. The recent merging of national-level institutions of water and spatial planning, such as in France [67] and the Netherlands [82], does not appear to be sufficient. For planners, flood management is seen as an overestimation of danger that could be solved technically to continue development when there is a low magnitude or low probability risk of flooding. Urban planners valorize the role of resilient urban planning measures,

such as upper-elevation, but without a comprehensive view of the potential indirect effects on flooding. Meanwhile, risk managers underestimate or misunderstand urban development opportunities [93]. In a central mode of governance based entirely on legal instruments, the way forward is to consider flooding from the point of view of urban design where planners propose local solutions that respect the general national rules. National authorities accept them—or not—and local authorities take legal responsibility for the project.

The second entry point, the recurrence and increase of damage caused by urban floods, is increasingly drawing attention to the links between development and prevention. A step forward can be taken in understanding governance issues better if literature looks specifically at the selection of instruments. This paper aims to boost debate on how planning and prevention could benefit from two-way interaction both by integrating flood planning into building, but also by changing the way building considers flooding. The objectives of spatial planning should be driven by public interest concerning environment and security, but the actual process of planning is a matter for legal experts, taking into account economic inputs. Legal, technical and scientific controversies underline that there is no overall agreement on the balance between development and prevention. Similarly, there is no political agreement between national and local authorities, between public authorities and civil society, between development and security perspectives. A complete comparative analysis of the selection of instruments in prevention, defense, mitigation, preparation and insurance strategies might examine in detail each of the following: the range of available options for prevention and the decisions made; the setting of agendas regarding instruments; and the implementation of measures, their evaluation and resulting modification.

**Funding:** This research received no external funding.

**Acknowledgments:** The paper was proofread by Anthony Cummins and re-read by Brynhild Drain-Brule.

**Conflicts of Interest:** The authors declare no conflict of interest.

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
