# Peer review of "What Can We Learn from Planning Instruments in Flood Prevention? Comparative Illustration to Highlight the Challenges of Governance in Europe"

_water, doi:10.3390/w12061841_

Round 1

Reviewer 1 Report

This study foucs on selection of planning instruments in flood prevention can be critical to gain a better understanding of governance. This choice is under-estimated in flood management literature. This paper fills a knowledge gap in flood management governance by examining the rationales for the choice of instruments. The paper is interesting and well displayed. In my opinion, the paper can be published after the language revised by a native English.

Author Response

We appreciate the reviewer comment. Thank you for allowing me to revise my paper. The advice of the peer reviewer has helped to improve my paper.

The rephrasing of the main subject of the paper will allow me to bring the theme more clearly.

Concerning the English language writing, I called upon a British proofreader and I will have another English-speaking colleague proofread it.

Reviewer 2 Report

The topic of the paper is interesting (and topical). After a short background story, which should be a bit more elaborate, and that suffers from some grammatical problems (row 47-49), it starts out really well with a part (2) about the role (and importance) of instruments for flood prevention, mainly in the form of spatial planning.

The methods section needs improvement. Mainly due to linguistic problems, but also because it is not clear why this method is chosen or why it is the most proper one for the study.

Part 4 (the comparative illustration) is interesting as it describes the three countries' flood prevention strategies. Some of the conclusions, e.g. on row 254 "instruments remains unsatisfactory", aren't really backed up by the study (or at all), which presents a sense of bias on behalf of the author. This is especially prominent from row 317 and onward: examples on row 318-320: "If the lack....", 347-352: conclusions are placed on top of each other, but the are not backed up by the study.

Part 5 seems to be about something almost completely different from the rest of the paper. This section is messy; it is difficult to know where the examples are taken from (which country). Row 375: "a never-ending story between..." what is this about? "Until 1980 the state" which state? Row 381: " when local authorities have more resources..." When is this, where is this? Row 398: "For planners..." Why? Because people still build in the areas or because there are still floods? Etc.

Specific comments:

  • The EU directive is usually referred to as the Floods directive.
  • Row 54: "the paper argues" => it is argued in the paper
  • Refrain from using it's, aren't etc.
  • If there is only one author to the paper, expressions like "our goal", "we propose" should perhaps be replaced.
  • Row 143: sentence is not finished? For how long?
  • Row 166: What does actors, rules and (especially) discourses mean in this context?
  • Row 181: There should not be a question mark at the end of the sentence.
  • Row 183: use "address" instead of answer (since there is no question in the paper, but rather an ambition to "look if the selection of ..."
  • Row 185-188: the meaning is lost due to grammatical errors.
  • Row 211: why a quote? This is the core of the paper...

Author Response

I thank the reviewer for the detailed consideration. The reviewer point furthermore highlights the importance of our avoiding subjective speculation on the role of instruments. I addressed the reviewer’ specific concerns point by point as follows :

  • Short background story needs to be more elaborate.

I have added some references on floods probability in a context of Climate Change, and on uncertainty in general. I also rephrase some sentences on diversification of strategies be more accurate. I add a sentence on gap knowledge and the focus of the article at the end.

  • The methods section needs improvement.

I have reworked the entire section, highlighting the role and need for the information collected. I explain further why I choose these three methodological steps and how I use them. I hope it is clearer now. Thanks for advice.

  • The comparative illustration needs to avoid subjective speculation

I have specifically re-written the conclusions on the countries' parties in an attempt to objectify the conclusions.  I think that I did not systematically give the references or theoretical bases to which I was making reference. For example, when I write that top-down regulatory instruments remain unsatisfactory it is with reference to the literature on participatory governance in risk management, which shows the limits, which are still strong today, on the involvement of the public. In the same way, I add some comment to explain why I say “Beyond the legal system, there is no global agreement on the balance between development and prevention” because I feel that you could have the same feeling of personal speculation.

Similarly, when I write “If the lack of inclusion of risk prevention in urban planning is offending flood managers” I have tried to make it clearer by endorsing my conclusion with studies on the analysis of administrative power relations between different public policy sectors.

  • Discussion section is messy and needs to be improved

This is undoubtedly the part I found the hardest to rework, but I have tried to re-organize the ideas, go back to the examples and get inspiration from the various feedback also received from the other reviewers. I hope that the re-writing of the text will help.

  • Grammatical problems should be solved point by point such as specific comments will be systematically rephrased and revised.

I'm will address the points of grammar, phrasing and clarity in a systematic way. Thank you for your careful reading. I would like to add that, at the end of the reading of the comments, I made a global re-reading of the text keeping in mind the general spirit of the comments, which led me to rephrase some sentences and add a few references. Finally, concerning the English language editing, I first called upon a British proofreader and I will have another English-speaking colleague proofread it again.

Reviewer 3 Report

The presented work aims to initiate a debate between the local-scale flood-management and the national-scale flood-prevention policies. The paper is of very good quality and presents the merits and demerits of both scales through the policies of three countries, namely England, France, and the Netherlands. The paper is well-written and the literature review is of good quality. I have only three minor suggestions:

1) Except for the mentioned advantages of local-scale flood management such as that "sometimes they have access to more precise data or they can pay for alternative experts", another merit is that the local authorities and have often better knowledge of the flood risk in their area. For example, a good practice when performing a hydrological-hydraulic study for an area is to ask the locals about the history of their area in terms of past floods events, specific locations that are usually flooded etc. This practice can be sometimes more revealing for the flood mitigation plan that any of the model outputs. In fact, the models should be able to reveal those flood-risk regions. However, often the studies are handled by non-local authorities that may or may not follow this practice, increasing the risk of poor prevention-mitigation measures.

2) Although I agree that "The objectives of spatial planning should be driven by the public interest, but the actual process of planning is a matter for legal experts, taking into account economic inputs.", I think that the planning process should be also take into account environmental and risk issues, and not only financial ones.

3) Another issue that is often neglected in flood management and prevention policies is the model uncertainties and the intrinsic uncertainty of the natural processes (e.g., Pappenberger et al., 2008; Dimitriadis et al., 2016; Teng et al., 2017). This uncertainty issue may contribute to the debate of the paper in two ways. First, since flood-risk and flood-planning can have large negative impacts on society, it could be dangerous for a local-authority to handle them and take responsibility for them, even if the local-authorities agree to such actions and accept any consequences. Second, the national-policies should always take into consideration any recent developments and the expertise of the local authorities and citizens. It is often the case that the national policies are based on Prevention Plans that have been conducted many decades ago, while the local-authorities are obliged to follow old-practices and traditional-models, even if scientific developments have proven those practices and models inadequate or even false. Therefore, local-authorities could handle more effectively flood risk and planning policies, since they are more easily adapted in the new research developments.

References

Dimitriadis, P., A. Tegos, A. Oikonomou, V. Pagana, A. Koukouvinos, N. Mamassis, D. Koutsoyiannis, and A. Efstratiadis, Comparative evaluation of 1D and quasi-2D hydraulic models based on benchmark and real-world applications for uncertainty assessment in flood mapping, Journal of Hydrology, 534, 478–492, 2016.

Pappenberger, F., K.J. Beven, M. Ratto, P. Matgen, Multi-method global sensitivity analysis of flood inundation models, Advances in Water Resources, 31 (1), 1-14, 2008.

Teng, J., A. Jakeman, J. Vaze,B.F.  Croke, D. Dutta, S. Kim, Flood inundation modeling: A review of methods, recent advances and uncertainty analysis, Environmental Modeling Software, 90, 201–216, 2017.

Author Response

I thank the reviewer for his insightful comments and recommendations. I have carefully considered the reviewer's suggestions and, in doing so, I feel that the manuscript is considerably strengthened. With respect to the three suggestions :

  • Thank you for mentioning the important scientific resource for local actors such as precise studies of small rivers which, when aggregated, give a more precise knowledge at the regional level. I will add this idea, thank you. I also agree with you to clarify this point: it is not always obvious that local stakeholders have access to hydraulic methods when they are contracted out to private consulting offices or insurance companies. Alos, they don't have the human and technical resources to combine or decompose these methods. I would be happy to discuss this further.
  • There is a great debate on the balance of interests between the public and private domains. Critical literature on this issue, particularly in the literature on planning and risk management, is not common. I'll reinforce that a little bit on your advice.
  • I fully agree that there is a lack of consideration for intrinsic uncertainty in the literature on flood planning and prevention and in practice as well. As regards the first idea on uncertainty and accountability, I have been working on this in urban flood zone project on another paper where I show how new legal instruments are being developed for the clarification of the balance of responsibilities. I should obviously say a word on this, thank you. On the second point, I can expand on the idea that local authorities are obliged to follow old practices and traditional models'. I think that the lack of capacity to manage flood risks more effectively is a real concern for local authorities. But in another way, it is also sometimes a piece in a wider controversy between development and prevention. I will try to be more specific on this point in the article. Thank you very much for the references I will insert in my article, and more generally in my research.

Round 2

Reviewer 2 Report

The improvement of the paper is substantial - all comments seem to have been addressed in this version, which thus can be accepted for publication in my opinion.

Author Response

Thank you for proofreading. The text has been re-read and corrected a second time to improve the English quality.